# Therapists' experiences with a new treatment combining physical exercise and dietary therapy (the PED-t) for eating disorders: an interview study in a randomised controlled trial at the Norwegian School of Sport Sciences

Maria Bakland,[1] Jorunn Sundgot-Borgen,[2] Rolf Wynn,[3] Jan H Rosenvinge,[4] Annett Victoria Stornæs,[2] Gunn Pettersen[1]

[1]Department of Health and Care Science, University of Tromsø, The Arctic University of Norway, Tromsø, Norway
[2]Department of Sports Medicine, Norwegian School of Sport Sciences, Oslo, Norway
[3]Department of Clinical Medicine, University of Tromsø, The Arctic University of Norway, Tromsø, Norway
[4]Department of Psychology, University of Tromsø, The Arctic University of Norway, Tromsø, Norway

**Correspondence to**
1 Maria Bakland;
maria.bakland@uit.no

## ABSTRACT

**Objectives** The aim of the current study is to explore how therapists running a guided physical exercise and dietary therapy programme (PED-t) experience their contribution to the treatment of patients with bulimia nervosa and binge eating disorder.

**Methods** Ten therapists running the PED-t were semistructurally interviewed and the transcribed interviews were analysed using a systematic text condensation approach.

**Setting** The study was run within the context of a randomised controlled trial at the Norwegian School of Sport Sciences.

**Results** The therapists experienced their knowledge about physical exercise and nutrition as important and useful, and that they could share their knowledge with the patients in different ways and with confidence in their own role. They also believed that their knowledge could serve as tools for the patients' post-treatment recovery and management of their daily lives. Moreover, the therapists put much effort in adjusting their teaching to fit each individual participant. Finally, they reported their personal qualities as important to build trust and therapeutic alliance.

**Conclusions** The terms 'clinical confidence' and 'alliance' may stand out as the overarching 'metacategories' covering the experiences revealed in this study. The clinical implication is that new groups of professionals may have an important role in the treatment of eating disorders.

**Trial registration number** NCT02079935; Results.

## Strengths and limitations of this study

► This study is the first reporting therapists' experiences with a specific treatment for eating disorders, and notably with a new group of professionals in the field.
► All therapists in the treatment programme were included in the study and were willing to participate in the study and share their experiences.
► A limitation with the study might be demand characteristic elicited by the recruitment procedure and the study context.

unfavourable longer-term course of illness.[1–3] A second challenge is to address the heterogeneity of BN and BED in treatments. Apart from core symptoms such as binge eating (BED and BN) and purging behaviours (BN), the heterogeneity relates to additional symptoms and complications of unhealthy nutritional and physical activity status, cognitive and regulatory processes, as well as experiential aspects related to suffering, the response to treatments and to the recovery process. The challenge then is to design effective treatments for BN and BED that address the heterogeneity of the disorders and the sufferers.

At present, however, the portfolio of evidence-based treatments for adults with BED and BN is virtually limited to cognitive–behavioural therapy (CBT).[4] Responding to the need to extend this portfolio to address the heterogeneity, a previous randomised controlled trial (RCT)[5] has shown that both nutritional therapy and guided physical exercise (PE) had equally beneficial effects as CBT in reducing bulimic symptoms. As a follow-up,

## INTRODUCTION

Treatment and recovery from binge eating disorder (BED) and bulimia nervosa (BN) may be challenging for therapists and patients. First, fragile motivation to change and the processual nature of recovery may conflict with the need for a rapid improvement of symptoms to lower the risk of an

a new RCT with an improved research design is in progress[6] where treatment effects of CBT are compared with a combination of physical exercise and dietary therapy (the PED-t programme) for patients with BN and BED.

Much is known from previous experiential research[7–14] about the level of patient satisfaction with various modes of treatment. However, the body of knowledge is limited about therapists' experiences. A few studies have shown some concordances between patients and therapists with respect to treatment qualities needed to ease recovery.[15 16] While such kind of studies have their merit in providing overall knowledge about treatments of eating disorders (EDs), there is a gap of knowledge about experiences from a therapist's perspective with a *particular* treatment programme, such as the new PED-t. How patients and their therapists experience different aspects of treatments is important to secure proper documentation of efficacy and effectiveness.

A focus on therapists' experiences with a particular treatment is important for at least three reasons. First, therapists' experiences with a particular treatment may provide a new and yet unexplored gateway to improve or adjust a treatment protocol or manual. Second, the therapists' experiences with a treatment and its procedures may determine the kind of clinical confidence that promotes the interaction of alliance and symptom change that may bring about a good treatment outcome.[17] A third, specific reason is related to the PED-t programme, where representatives from uncommon professions, that is, dieticians and specialists in PE, are the therapists.

The aim of this study is to explore the therapists' experiences of contributing in a new treatment for patients with BN and BED—that is, PED-t.

## METHODS
### Study context
The current study was conducted within the context of an RCT.[6] This trial's aim was to test whether the combination of physical activity and dietary therapy was more effective in treating EDs than CBT. Included in this RCT were women aged 18–40 years with a Diagnostic and Statistical Manual of Mental Disorders-5 (DSM-5) BN or BED for at least 3 months prior to the current study, and who had no major comorbid disorders.[6] Both treatments were run within a group format of 20 sessions over 16 weeks and each group consisted of five to seven participants. Ten therapists were engaged to run the PE and nutritional treatment groups (PED-T programme).

### Participants and recruitment
All therapists (n=10) engaged in the RCT, having finished one or more treatment groups, comprised the sample of the present study. They were 25–35 years of age and with a professional background as physical trainers or dietitians, holding a master or a bachelor degree in sport sciences, PE or exercise medicine, and with at least 3 years of practice with supervised exercise.

The principal investigator of the RCT (JS-B) contacted all therapists with a request to participate. As the therapists responded positively to participate in the study, the first author (MB) contacted them and made appointments for the interviews.

### Interview procedure
The first author conducted the interviews between September 2016 and January 2017. Six of the interviews took place at the Norwegian School of Sport Sciences where the RCT was running. The remaining four interviews took place at other locations of the informants' own choice. The interviews lasted for about 1 hour. Using a semistructured interview guide, the interviews were conversations about the therapists' experiences of their own contribution in the RCT treatment. All interviews were recorded and transcribed verbatim by the first author. The participants received a gift card with the value of £25 for their participation.

### Data analysis
The data were analysed inspired by the principles of systematic text condensation (STC).[18] The STC analysis is organised as a four-step approach. The first step comprised reading the transcribed interviews thoroughly to achieve an overview of the data and to approach first-impression themes. The transcripts were also compared with the audiotaped interviews to check the accuracy of the text. In the second step, the first author identified and coded units of meaning related to the therapists' experiences of their contributions to the trial, and V.10 of the NVivio software was used to document this step. The analysis then continued with the third step where the coded data were condensed and meanings were abstracted within each of the categories and subcategories and quotes were selected to illustrate the meaning of each group. In the final step, the content within each group was synthesised and renarrated.[18] The first author (MB) who has a background as a mental health nurse with knowledge and experience in the field conducted the analysis, while additional authors validated the process of STC and discussed the interpretations until agreement was reached. The findings were also subsequently discussed in the author group, and the final descriptions were a result of a (hermeneutical) process moving back and forth between the transcripts, the findings, the literature and relevant theory, to secure that the constructed descriptions were grounded in the empirical data.[18] The analysis resulted in two main categories: contributing with knowledge and contributing with personal qualities.

Table 1 gives examples of first-impression themes, meaning units, codes, code groups, condensed units and descriptions.

### Ethical considerations and data security
The therapists signed an informed consent form to participate, and they were informed about the possibility that they could withdraw from the study at any time, without

**Table 1** Examples of first-impression themes, meaning units, codes, code groups, condensed units and descriptions derived from the data analysis

| Step 1: total impression (from chaos to themes) | Step 2 | | | Step 3: condensed unit | Step 4: description |
|---|---|---|---|---|---|
| | Meaning units | Code | Code group | | |
| To know one's own limitations | "I can give advice, as a compassionate fellow human, but I do not have sufficient knowledge to give the conversation a psychologist would give." | To contribute with knowledge | To know one's own limitations | It is sometimes challenging to know what to say | Sometimes feeling a lack of knowledge, wanting to not give misleading information |
| Trust | "You have to express care as well, for them to trust you. The trust is important." | To contribute with personal qualities | Building trust | The trust is important and not everyone experience it from the beginning | Underlining the importance of mutual trust |

giving any reason. Data were treated confidentially and information about the therapists was presented in a way so that they were not identifiable. All transcripts were deidentified and pseudonyms were used.

## User involvement in the study
Prior to the data collection, the focus and aim of the study were discussed in cooperation with coresearchers who are members from a local patient organisation on EDs. User involvement is anticipated to lead to more relevant research questions, more accurate data and research findings that are more likely to have influence.[19] The coresearchers have been a part of the project, including the current study, from the beginning and will continue throughout the project and the next two studies. In addition to discussing focus and aim of all three studies, the coresearchers have been part of preparing interview guides and information sheets. In regular meetings throughout the project, they have also been good discussion partners and have shared valuable personal experiences about ED.

## RESULTS
### To contribute with knowledge
This main category concerns the physical trainers and dietitians' experiences of contributing in the treatment programme with knowledge about PE and nutrition. They described being able to share their knowledge with the women with EDs in different ways and with confidence in their own role. The category is described through three subgroups: 'Giving away tools by teaching', 'A balancing act' and 'To know one's own limitations'.

### Giving away tools by teaching
The therapists reported that their main responsibility in the project was to teach women with EDs how to exercise and eat in a normal and healthy way, in order to correct misinterpretations and misunderstandings about physical activity, food and nutrition.

They expressed that their own knowledge and competence in teaching exercise and nutrition was a result of their education as well as their experiences through work and daily life. Furthermore, they experienced that this knowledge made them able to teach the women why and how to exercise and eat normally.

Meghan, a physical trainer, described how she experienced teaching:

I believe that is what helps them, mainly because they get to know their own body. How the body reacts after receiving the right amount of food after exercise. How the body feels like after 16 weeks of working out. That the body is functioning in a different way than it used to. That they feel like this is the way they want their body to function.

According to the therapists' opinion, the women's own experiences of the impact of adopting healthier lifestyles and physical activity habits were the best way for the women to learn. However, one PE trainer also mentioned that some of the women might not have been ready to implement this knowledge in their daily lives at that point. Nevertheless, teaching was experienced as important by the therapists because the knowledge might function as tools that the women could use to manage their daily life in the future. Ben, who is also a physical trainer, explained how he trusted his own knowledge and how this gave him confidence in his role:

I trust that knowledge, and especially because it comes from The Norwegian School of Sport Science. I often have great confidence in what I know, and that is important, because then you experience confidence in your own role, and I believe they can tell if you have that.

Ben talked about misinterpretations and described the tools they provided to the participants like this:

You learn some tools when you participate in this treatment. Some tools that you can use, and that you have been able to try out and get feedback on. It is about what to eat and how to exercise in a smart way. You have some of your misinterpretations corrected. That is knowledge that you can carry with you for the rest of your life.

The therapists thus experienced sharing knowledge that could help the women with EDs managing their daily lives after treatment as an important contribution.

## A balancing act

The therapists reported being aware of individual differences among the participants and that they put much effort in adjusting their teaching to fit each individual woman. Susan, a physical trainer, described the women's vulnerability:

> It is so important though, because they are in such a vulnerable situation, that it might become overwhelming if we focus too much on exercise and food. (…) So it is always a balancing act.

Thus, the physical trainers needed to adjust the exercises, but more importantly, all of the therapists reported the need to adjust the knowledge level and focus in their teaching.

To keep the right balance between to 'push' and to 'hold back' was perceived as difficult at times. According to the therapists, some of the women needed to be 'pushed' a little to be able to lift the right amount of weights, or finding their place in the groups. However, the therapists simultaneously described how one or several women could have a bad day or feel unwell, and in these cases, pushing the participants would be wrong. Meghan, a physical trainer, said it in this way:

> We need to find the balance, when to push and when to hold back. It is not suitable to push at all times, in a way you need to adjust to what kind of day they are having.

Knowing how much and when to praise the women was also described as important by the therapists. They reported that for some women, it was important to be worthy of praise, but at the same time this could make the next day for them difficult if they did not manage in the same way. Furthermore, the therapists experienced that some of the women might feel worse when somebody else was praised. Starting each group session talking to the women and sensing how they felt made the therapists being able to adjust to each individual's needs. Kirby, who is a dietitian, described the importance of being open and controlling at the same time:

> We should be open to meet the participant where they are, but at the same time we have a plan according to protocol, and therefore we have to be somewhat controlling in the conversation.

Altogether, the therapists found it challenging to follow the protocol and simultaneously make individual adjustments. However, they found themselves being able to balance each individual woman's need, and this was perceived as an important contribution.

## To know one's own limitations

The therapists described that the participants asked many questions, to secure an optimal outcome and use of time in the treatment programme. At times, when the therapists reported being unable to answer, they felt a considerable responsibility of not giving misleading information. Thus, they stressed the importance of bringing such unanswered questions forward, by asking their superiors, in order to provide a good answer on the next occasion.

A minority of the therapists reported having experienced a situation where they felt a lack of knowledge in mental health disorders. Susan, a physical trainer, said it like this:

> I can give advice, as a compassionate fellow human, but I do not have sufficient knowledge to give the conversation a psychologist would give.

In these cases, they experienced the participants as having a need for attention and to be listened to. The therapists also reported being aware of the potential that the participants were having a difficult time outside the treatment context.

In general, most of the therapists could give an example where they had to acknowledge not having sufficient knowledge. Kirby, a dietitian, reflected about a situation where he felt not able to help:

> So I guess I couldn't help her in that situation, but at the same time I tried to tell her that I couldn't help and I tried to show some compassion or understanding. Yes.

Louise, also a physical trainer, expressed the feeling of responsibility:

> Perhaps I could have produced a logical answer if it was to a friend of me, but these are women that have a disease and they deserve an answer that is good enough.

Despite not having the sufficient competence in all situations, the therapists described an insight in their own limitations. However, they reported a feeling of substantial contribution in general because of their strive to always meet the participants' requests.

## To contribute with personal qualities

The therapists highlighted the importance of having certain personal qualities themselves. Teaching the women with EDs was perceived not only as a transference of knowledge, but needed to be done with respect, empathy, interest and joy. This category constitutes three subgroups: 'Being a fellow human being', 'Building trust' and 'Creating a good group dynamic'.

## Being a fellow human being

The therapists reported the importance of having different qualities when working together in a team, which contributed in providing many forms of help and assistance. Kim, a physical trainer, said:

> I believe the fact that we instructors are different as persons is a major advantage. When we are working together, it helps having different qualities,

complementing each other. We received positive feedback regarding that.

Being compassionate, understanding and having a desire to help were highly valued qualities that the therapists reported having acquired through practical professional experience, and which had influenced their way of working with the participants.

Most of the therapists provided examples where they had to give the women something more than just the regular follow-up according to the protocol. When one of the women had a difficult time and rushed out of the group session, the therapist had to follow her out in the hallway, be a good discussion partner and try to make her overcome her difficulties. The therapists reported that these conversations in the hallway, being a fellow human being, most often were sufficient to get the women to return to the group sessions. Louise, who is a physical trainer, described how she used herself as a fellow human being. In the situation below, 'she' is a woman with ED:

> Sometimes the important thing is that she feels good when she goes home. Just leave the group, take your time and tell the other therapists that you have to talk to her. Make her understand that you too, have difficult times, even though you're not sick or in a bad shape. Finally, she is able to return to the group session, and you can see, it is as if she is glowing in joy! Then you feel good because you made her day and you put out a small fire.

Emmely, a physical trainer, said:

> Trying to get to know them, showing them that you care and have an interest in them as persons. That this isn't just research. Being a fellow human, yes.

Being a fellow human being was among other things described to be an important contribution when it came to helping the participants overcoming challenges.

### Building trust
The therapists experienced that the treatment required a mutual trust between themselves and the group members. From some of the women, such a trust came rather immediately, while other women needed more time, and the therapists outlined the need to work throughout the process to establish and maintain this trust. Moreover, the therapists described having used several ways of getting acquainted, and building and maintaining trust.

The physical trainers experienced the building of trust as essential in the sense that they reported the women had to rely on them, placing the right amount of kilos on the weights during the strength training. According to the therapists, the weights should not be too heavy for the women to manage, but conversely, not be too easy to promote progression.

Sometimes the therapists experienced being tested whether they were coordinated and whether they knew what they were doing. For example, a therapist could be asked a question and then another therapist was asked the same question. To avoid splitting, the therapists thus experienced it as helpful to have regular meetings between themselves beyond what was planned in the protocol. In these meetings other challenges and current needs were also discussed.

In the interviews, the therapists also underlined the importance of mutual trust. This involved the therapists showing the women that they had expectations regarding attendance and involvement. By expecting the women to show up, they reported an experience of being helpful to the women to feel committed to something. The therapists reported a feeling that this commitment might be what lead the woman to show up at practice even if they were having a bad day and wanted to stay at home.

Susan, a physical trainer, described building trust like this:

> Of Course, we have to be clear and professional. However, that is not enough. You have to express care as well, for them to trust you. The trust is important.

Tabby, also a physical trainer, described the support a woman needed from her like this:

> One of the women wanted me to stand beside her screaming, because she knew that she worked better if I pushed her.

Andrea, who is a dietitian, experienced that being open and direct herself helped the women:

> In this treatment-program we discuss openly what happens to your body when you vomit or use laxatives or things like that. I feel that this helps the women to be more open in return, telling us about their experiences.

The importance of the mutual trust was described in different ways but represented evenly among all the ten therapists.

### Creating a good group dynamic
The therapists described the groups as very different from each other in their way of functioning. Some groups were well established rather immediately, resulting in a good environment where almost every session was perceived as mutually rewarding and fulfilling. Other groups had a poorer start resulting in a lot of absence, maybe even dropouts.

To contribute in making a good group, the therapists reported being aware of the need to facilitate getting to know one another and having fun together. The physical trainers would sometimes get help from a woman coping with an exercise to help another group participant. However, they also reported being aware of the possibility that this would lead someone coping less well to feel even worse. The physical trainers also described working against groupings by varying who exercised together each time. In general, all the therapists reported working in a manner that would leave a space for every woman in

the groups. Working together as a team, the therapists emphasised the importance of giving each other time to follow the women out into the hallway to talk and comfort when they had a tough time.

Andrea, a dietitian, explained how she contributed to making a good group:

We try to give all the participants the space they need to talk, and sometimes we even have to make them.

Tracey, also a dietitian, described it like this:

It is challenging when we are working with groups, and some of them might have stronger personalities than others. Other women might even dislike them. As a therapist, you have to balance this, or make everyone accept that this is how it is.

All together, working towards good group dynamics was experienced as an important contribution.

## DISCUSSION

The aim of this study has been to explore therapists' experiences of their contributions in providing a novel combination of PE and dietary therapy (the PED-t programme) for ED. The results showed that the PED-t therapists experienced that their professional knowledge about PE and nutrition was an important source of patients' trust in the PED-t programme.

Two main findings from this study may serve as indices of therapists' trust and confidence in their professional skills. First, the PED-t therapists reflected around the use of the knowledge they held, and focused on how this knowledge might serve as tools in the future recovery process for the patients. Indeed, it takes confidence to try to use own knowledge to provide others with transferrable skills beyond the original local constellation or context and beyond the mere purpose of adhering to a protocol. Moreover, the PED-t therapists regarded themselves as able to evaluate what knowledge was most important at any time. For instance, it was at times perceived as more important to ensure that the participants left a treatment session feeling good about themselves than always following the protocol. From a research point of view, to divert from the protocol is not very recommendable. Obviously, confidence is not a matter of notoriously breaching a protocol, but is rather a matter of profound knowledge and understanding of the general principles of the protocol, practical skills in implementing it, and the freedom to make temporary adjustments to accommodate for the immediate needs of the patient. In the general psychotherapy research, there is mixed evidence for the importance of confidence to treatment outcome.[20] However, in the field of ED, confidence has been found to be central to patients' positive experience of treatment success.[21] Knowledge is a prerequisite for confidence, yet confidence indicates the ability to flexibly use one's knowledge.

Second, and in contrast to previous research,[16] where therapists valued the focus on changing the behavioural symptoms of ED, this study shows that the PED-t therapists understood their personal qualities as fellow human beings as important for the progress of the participants. Personal qualities were manifested as a focus on creating a good group dynamic and a positive atmosphere through directly communicating empathy, interest and care. The mentioned personal qualities are also among the core qualities of alliance and a good relation between patients and therapists. The fact that the therapists in this study reported being empathic, interested and caring, and that they had no previous experience in treating ED patients, stands in contrast to previous findings[22] showing that especially less-experienced clinicians tend to show negative reactions towards these kind of patients.

Our findings then align with psychotherapy research,[20 23] highlighting the importance of a strong alliance for a good treatment outcome. Within the field of EDs, a meta-analysis[17] shows a reciprocal relation between alliance and symptom change. The current study does not report on symptom change. However, one may at least expect that such change have been promoted by the fact that the therapists reported having been focused on providing tools for recovery as well as building trust between themselves and the group members. Thus, our findings indicate that the PED-t may be an excellent arena for building alliance, as the therapists were able to 'small talk' during the PE sessions and still with a mutual focus on the PEs. Awaiting experiential evidence from the participants, a high correspondence between therapists' focus and participants' priorities might be the case in the PED-t program. Forthcoming research including outcome measures of the intervention may elucidate whether such a possible correspondence is impeding or facilitating treatment progress and recovery.

A clinical implication from the therapists' experiences is that dietitians and specialists in PE can contribute in running a treatment programme for patients with ED. This aligns with dissemination trials with CBT.[24–27] Hence, what is important seems not to have a particular professional background, but to have specific knowledge and understanding of the therapeutic procedures and the confidence to implement them in a clinical setting. Finally, the therapists also provided important experiences and ideas on how to redefine and adjust the PED-t programme. For instance, further to strengthen their confidence to the benefit of the patients, regular supervision and debriefing should be introduced. By own initiative, the PED-t therapists met on a regular basis throughout the treatment programme to reflect around their own contribution and potential challenges. Such meetings should be obligatory in a future implementation of the programme. In addition, the selection procedure might be slightly revised to provide more information about the purpose and procedures of the PED-t. On the other hand, the therapists expressed the importance of continuing including only patients with mild to severe

symptoms and not those with other, concurrent mental disorders who obviously might benefit from treatments not focusing on the ED. An obvious strength of the present study is the originality in reporting therapists' experiences with a specific treatment of ED. In a study under way, we will explore the experiences of the participating patients. Possible demand characteristic elicited by the recruitment procedure and the study context may have inflated the overall positive experiences reported by the therapists. However, such a limitation seems rather unlikely because diverse opinions and experiences were reported. For this reason, the present results may be transferrable to other clinical and non-clinical contexts. Hence, along with ED-relevant outcome measures, the experiential data may contribute to an evidence base for how the PED-t might be disseminated in new contexts, and possibly catching many sufferers who do not appear as patients in the healthcare system.

**Acknowledgements** The authors would like to thank the Norwegian Women Health organisation who founded the RCT.

**Contributors** MB, RW, JHR, GP and JS-B were all responsible for the planning of the study. MB collected and analysed the data, and the results were validated by GP, RW and JHR. All authors, MB, RW, AVS, RW, JHR, GP and JS-B, contributed in finalising the manuscript.

**Funding** The publication charges for this article have been funded by a grant from the publication fund of University of Tromsø, The Arctic University of Norway.

**Competing interests** None declared.

**Ethics approval** The study was approved by the Regional Committee for Medical and Health Research Ethics identifier 2013/1871 on 23 October 2013 and prospectively registered in ClinicalTrials.gov identifier NCT02079935 on 17 February 2014.

**Provenance and peer review** Not commissioned; externally peer reviewed.

**Data sharing statement** No additional data are available.

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
