## [Reviewer comments · BMJ Open]

ARTICLE DETAILS

TITLE (PROVISIONAL)	Therapists' experiences with a new treatment combining physical exercise and dietary therapy (the PED-t) for eating disorders: An interview study in a randomized controlled trial at the Norwegian School of Sport Sciences.
AUTHORS	Bakland, Maria; Sundgot-Borgen, Jorunn; Wynn, Rolf; Rosenvinge, Jan; Stornæs, Annett; Pettersen, Gunn

VERSION 1 – REVIEW

REVIEWER	Sarah Fogarty Western Sydney University, Australia
REVIEW RETURNED	30-Sep-2017

GENERAL COMMENTS	The paper is very well written and presents a novel area of research and treatment.
---

REVIEWER	Brian Cook California State University Monterey Bay
REVIEW RETURNED	25-Oct-2017

GENERAL COMMENTS	This appears to be the same manuscript that I have already reviewed in it's initial submission and again in it's revised submission. I had previously indicated that the revise version had addressed my previous concerns and was acceptable for publication.
--

REVIEWER	Ulrich Schweiger Lübeck University Medical School, Germany
REVIEW RETURNED	11-Dec-2017

GENERAL COMMENTS	This is an interesting report on the experience of physical trainers and dietitians within an innovative treatment for bulimia nervosa and binge eating disorder. It is sound with respect to methodology and presentation. It would be interesting to link the propositions of the therapists to the experience of the patients, but this may be beyond the scope of this paper.
---

VERSION 1 – AUTHOR RESPONDS

The minor revision according to the formatting has now been done.
The request from reviewer #3 to include the experiences of the participants is pertinent, but these aspects will be covered in a paper under preparation.

With this in mind, we do hope that the manuscript now is satisfactory, and we look forward to your final decision about publication in the BMJ Open.